# Dysbiotic oral microbiota and infected salivary glands in Sjögren's syndrome

**Jehan Alam**[1,¤a], **Ahreum Lee**[1], **Junho Lee**[1], **Dong Il Kwon**[1,¤b], **Hee Kyung Park**[2], **Jung-Hyun Park**[3], **Sumin Jeon**[1], **Keumjin Baek**[1], **Jennifer Lee**[4], **Sung-Hwan Park**[4], **Youngnim Choi**[1]*

**1** Departments of Immunology and Molecular Microbiology, Seoul National University School of Dentistry, Seoul, Korea, **2** Departments of Oral Medicine and Oral Diagnosis, School of Dentistry and Dental Research Institute, Seoul National University, Seoul, Korea, **3** Experimental Immunology Branch, National Cancer Institute, NIH, Bethesda, Maryland, United States of America, **4** Division of Rheumatology, Internal medicine, Seoul St. Mary's Hospital, The Catholic University of Korea, Seoul, Korea

☯ These authors contributed equally to this work.
¤a Current address: Department of Ophthalmology, Baylor College of Medicine, Houston, Texas, United States of America
¤b Current address: Division of Integrative Biosciences and Biotechnology, Pohang University of Science and Technology, Pohang, Korea
* youngnim@snu.ac.kr

**Data Availability Statement:** The sequence data are available in the NCBI SRA database (SRP158992).

**Funding:** This study was supported by a grant [HI13C0016] awarded to Youngnim Choi from the Korea Health Industry Development Institute and a

## Abstract

Key events in the pathogenesis of Sjögren syndrome (SS) include the change of salivary gland epithelial cells into antigen-presenting cell-like phenotypes and focal lymphocytic siala-denitis (FLS). However, what triggers these features in SS is unknown. Dysbiosis of the gut and oral microbiomes is a potential environmental factor in SS, but its connection to the etio-pathogenesis of SS remains unclear. This study aimed to characterize the oral microbiota in SS and to investigate its potential role in the pathogenesis of SS. Oral bacterial communities were collected by whole mouthwash from control subjects (14 without oral dryness and 11 with dryness) and primary SS patients (8 without oral dryness and 17 with dryness) and were analyzed by pyrosequencing. The SS oral microbiota was characterized by an increased bacterial load and Shannon diversity. Through comparisons of control and SS in combined samples and then separately in non-dry and dry conditions, SS-associated taxa independent of dryness were identified. Three SS-associated species and 2 control species were selected and used to challenge human submandibular gland tumor (HSG) cells. Among the selected SS-associated bacterial species, *Prevotella melaninogenica* uniquely upregulated the expression of MHC molecules, CD80, and IFNλ in HSG cells. Concomitantly, *P. melaninogenica* efficiently invaded HSG cells. Sections of labial salivary gland (LSG) biopsies from 8 non-SS subjects and 15 SS patients were subjected to *in situ* hybridization using universal and *P. melaninogenica*-specific probes. Ductal cells and the areas of infiltration were heavily infected with bacteria in the LSGs with FLS. Collectively, dysbiotic oral microbiota may initiate the deregulation of SGECs and the IFN signature through bacterial invasion into ductal cells. These findings may provide new insights into the etiopathogenesis of SS.

grant [2016R1E1A1A01942402] awarded to Youngnim Choi from the National Research Foundation of Korea. Labial salivary gland biopsy specimens used in this manuscript are from the Sjögren's International Collaborative Clinical Alliance [SICCA], funded under contract N01 DE-32636 by the National Institute of Dental and Craniofacial Research, with funding support from the National Eye Institute and the Office for Research in Women's Health. The funders had no role in study design, data collection and analysis, decision to publish, or preparation of the manuscript.

**Competing interests:** The authors have declared that no competing interests exist.

## Introduction

Sjögren syndrome (SS) is an autoimmune disorder characterized by dryness of the mouth and eyes. Focal lymphocytic sialadenitis (FLS) is one of the diagnostic criteria for SS and manifests as mononuclear cell infiltrates consisting of lymphocytes, macrophages, follicular dendritic cells, and dendritic cells [1, 2]. In early lesions, the infiltrate consists mostly of T cells and tends to form around the ducts; in advanced lesions, on the other hand, B cells predominate, and the infiltrate extends to acini as well [3]. FLS with a focus score of ≥1 is strongly associated with serum anti-SSA/SSB positivity, hypergammaglobulinemia, abnormal ocular surface staining, and abnormal unstimulated whole salivary flow in SS patients [4]. In the mouse models of SS, the occurrence of FLS precedes the decrease in salivary flow by 4 to 8 weeks [5]. Therefore, FLS is a key event in the pathogenesis of SS.

At the cellular level, SS is often described as an "autoimmune epithelitis". Salivary gland epithelial cells (SGECs) in SS patients often present the phenotype of antigen-presenting cells (APCs) by expressing class I and II MHC molecules, costimulatory molecules CD80 and CD86, and adhesion molecules, such as ICAM-1 and VCAM. Thus, SGECs in SS patients are equipped with surface molecules necessary for T cell activation and epithelial-T cell synapse formation [6]. In addition, SGECs in SS patients produce chemokines that mediate lymphocytic infiltration and the formation of ectopic germinal center (GC)-like structures [7, 8]. Thus, the epithelitis seems to be involved in the initiation and perpetuation of the lesions. However, what triggers the epithelitis in SS has remained unclear.

A growing body of evidence suggests that dysbiosis of the oral microbiome is associated with the pathogenesis of several autoimmune diseases, such as systemic lupus erythematosus, Crohn's disease, and rheumatoid arthritis. [9]. Dysbiosis of the oral microbiota in SS has been reported [10–16], but its connection to the etiopathogenesis of SS remains unclear. In addition, a recent study reported that the salivary microbiota of SS patients is comparable to that of non-SS sicca patients [17]. This study aimed to characterize the oral microbiota in SS and to investigate its potential role in the pathogenesis of SS through additional *in vitro* and *ex vivo* studies using selected bacterial species.

## Materials and methods

### Study ethics

This study was carried out in accordance with the Declaration of Helsinki. The protocol was approved by the Institutional Review Boards at Seoul St. Mary's Hospital (KC13ONMI0646) and at Seoul National University, School of Dentistry (S-D20140022, S-D20170004). All subjects gave written informed consent in accordance with the Bioethics and Safety Act in the Republic of Korea.

### Human samples

The SS patients were diagnosed at the Department of Rheumatology Seoul St. Mary's hospital between October 2013 and April 2014 by the 2002 American-European Consensus group classification criteria [18]. Patients with drug-related dry mouth were diagnosed at the Department of Oral Medicine, Seoul National University Dental Hospital. The list of medication is summarized in S1 Table. Healthy individuals had no subjective oral dryness or other discomfort in the oral cavity and were on no medications. Exclusion criteria included age under 20, smoking, and the use of antibiotics, steroid, or immunosuppressant within a month prior.

For bacterial sampling, subjects were asked to avoid eating and antiseptic mouthwashes for two hours before sampling. Oral bacteria were collected by vigorous gargling of 5 ml distilled

water for 30 seconds from control subjects, including 15 healthy individuals and 10 drug-related sicca patients, and from primary SS patients, including 8 without dry mouth and 17 with dry mouth defined by unstimulated whole salivary flow rate (UWSFR) ≤ 0.1 ml/min. One of the healthy subjects had dry mouth despite the lack of any diseases or medication. All subjects were female because male SS patients are very rare in Korea. The sample size required for this study was determined as at least 5 per group based on the previous study, which reported the inter-group distance for tongue microbiota between SS and healthy individuals as 0.382 [11], and simulated PERMANOVA power estimation presented by Kelly *et al.* [19].

The sections of paraffin-embedded labial salivary gland (LSG) biopsies with various FLS from 15 SS patients (mean age: 56.1 ± 3.5) and 8 non-SS subjects (mean age: 43.5 ± 4.4) excluded from diagnosis with SS were obtained from the Sjögren's International Collaborative Clinical Alliance (The University of California, San Francisco, CA, USA). All biopsies were from Asian female.

## Oral microbiota analysis

DNA was extracted from whole mouthwash pellets by first agitating with garnet beads (MO BIO Laboratories, Carlsbad, CA, USA) for 10 minutes and then subjected to bacterial genomic DNA extraction kit (Thermo Scientific Korea, Seoul, Korea). Extracted DNA was amplified with primers targeting the V1 to V3 hypervariable regions of the bacterial 16S rRNA gene, and the amplicons were sequenced using a 454 GS FLX titanium pyrosequencer (Roche, Branford, CT, USA) and analyzed at Chunlab (Chunlab Inc., Seoul, Korea) as previously described [20]. The taxonomic classification of each read was assigned against the EzTaxon-e database (http://eztaxon-e.ezbiocloud.net). The alpha diversities were determined by Chao1 and Shannon diversity indices at the 3% distance. The overall phylogenetic distance between communities was estimated using the weighted Fast UniFrac with normalization for the read size and was visualized using principal coordinate analysis (PCoA). The sequence data are available in the NCBI SRA database (SRP158992).

## Quantitative PCR

To quantify the bacterial loads of oral microbiota, real-time PCR was performed using SYBR$^®$ Premix Ex Taq (Takara, Kusatsu, Shiga, Japan) and a pair of primers targeting the conserved region of the bacterial 16S rRNA gene: 5'-AGTCACTGACGAGTTTGATCMTGGCTCAG-3' and 5'-CAGTGACTACWTTACCGCGGCTGCTGG-3'. The copy number of the 16S rRNA gene was calculated using standard curves generated by parallel amplification of the genomic DNA of *Porphyromonas gingivalis* ATCC 33277.

## *In vitro* challenge of human submandibular gland tumor (HSG) cells with bacteria

Three SS-associated and 2 control bacterial species were selected and used to challenge HSG cells. Bacteria used in this experiment were obtained from Korean Collection for Type Culture (KCTC, Jeongeup, Jeollabuk-do, Korea) and the American Type Culture Collection (ATCC, Manassas, VA, USA). *Prevotella melaninogenica* KCTC 5457 and *Prevotella histicola* KCTC 15171 were cultured in KCTC-5457 medium and KCOM3 medium, respectively. *Fusobacterium nucleatum* ATCC 25586, *Streptococcus salivarius* KCTC 5512, and *Rothia mucilaginosa* KCTC 19862 were grown in BHI medium supplemented with 5 μg/ml of hemin and 10 μg/ml of vitamin K. All bacteria were cultured under anaerobic conditions (5% $H_2$, 10% $CO_2$, and 85% $N_2$) at 37°C, harvested in log phase, and washed with PBS before use.

HSG cells, a transformed human submandibular gland intercalated duct cell line [21], were maintained in DMEM complemented with 10% fetal bovine serum and 100 unit/ml penicillin and streptomycin at 37˚C in a water-saturated atmosphere of 95% air and 5% $CO_2$.

HSG cells ($4 \times 10^4$ cells/well) were seeded into 24-well plates in antibiotic-free medium one day before bacterial challenge. At 70% confluence, the HSG cells were cocultured with each bacterial species at a multiplicity of infection (MOI) of 50 and 100 for 3 days in the absence or presence of 100 ng/ml IFNγ (PeproTech Korea, Seoul, Korea). To prevent the outgrowth of bacteria, gentamicin was added to the medium 6 hours after bacterial infection. Three to four bacterial species were challenged at a time, and the results of 3 experiments were pooled.

## Analysis of MHC and costimulatory molecules expressed on HSG cells

HSG cells challenged with bacteria were stained with FITC-conjugated anti-human HLA-DR, DP, DQ monoclonal antibody (mAb) clone Tu39 (BD Bioscience, San Diego, CA, USA), PE-conjugated anti-human CD80 mAb clone L307.4 (BD Biosciences), PerCP-conjugated anti-human HLA-A, B, C mAb clone W6/32 (BioLegend, San Diego, CA, USA), and APC-conjugated anti-human CD86 mAb clone IT2.2 (BioLegend), and then analyzed using a FACSCalibur flow cytometer (BD Biosciences).

## Measurement of IFNλ produced by HSG cells

The amounts of IFNλ1/3 in the culture supernatant of HSG cells challenged with bacteria were measured using an ELISA kit (R&D System, Minneapolis, MN, USA).

## Measurement of bacterial invasion into HSG cells

HSG cells were seeded onto 12 mm coverslips one day before infection. HSG cells were infected with each bacterial species stained with both pHrodo Red succinimidyl ester (Invitrogen, Carlsbad, CA, USA) and carboxyfluorescein succinimidyl ester (CFSE) at MOI 100 for 6 hours. HSG cells infected with double-stained bacteria were examined by confocal microscopy (Carl Zeiss, Jena, Germany) after staining with Hoechst 33342.

## *In situ* hybridization (ISH) of bacterial 16S rRNA

Preparation of a universal probe and ISH were performed according to a previous report [22, 23]. A *P. melaninogenica*-specific probe was prepared by PCR amplification of *P. melaninogenica* gDNA using a pair of primers 5′-TCTGAACCAGCCAAGTAGCG-3′; and 5′-TACACGACGAA TTCCGCCAA-3′ and labeling with digoxigenin. The specificity for *P. melaninogenica* was confirmed by dot blotting using various bacterial lysates (S1 Fig). Sections of LSG biopsies from 8 non-SS subjects and 15 SS patients were subjected to ISH using universal and *P. melaninogenica*-specific probes. Three areas with strong signals were preferentially photographed at x400 magnification without knowing the disease status, and the intensities of ISH signals were measured using the color deconvolution plugin of ImageJ (National Institute of Mental Health, Bethesda, MD, USA).

## Statistics

The t-test was used to determine differences between control cells and cells challenged with bacteria. Nonparametric methods, the Mann-Whitney U, Kruskal-Wallis followed by post hoc with Bonferroni adjustment, and Spearman's rank correlation tests, were used to analyze the data from *ex vivo* experiments using tissue sections and bacterial samples unless stated otherwise. To identify SS-associated species, a logistic regression analysis and linear discriminant

analysis (LDA) were used. The LDA was performed with the LDA effect size (LEfSe), using an alpha value of 0.05 for the Kruskal-Wallis test and a threshold of 2.5 for the logarithmic LDA score [24]. Benjamini–Hochberg test for a false discovery rate of 0.2 was also performed. Difference in bacterial communities between groups was determined by the PERMANOVA test using the R vegan package. The other statistical analyses were performed using SPSS Statistics 22 software (SPSS, Inc., Chicago, IL, USA). Significance was set at $P < 0.05$.

## Results

### Dysbiosis of the oral microbiota in SS patients

To characterize SS-associated changes in oral microbiota, excluding the effect of oral dryness, we compared control and SS groups and their subgroups, stratified by the absence or presence of dryness. The characteristics of the groups and subgroups are summarized in Table 1. According to estimated bacterial loads, the SS group had a significantly higher total bacterial load than the control group, which was evident in a non-dry condition (Fig 1A). While the species richness estimated by Chao1 did not differ between the groups, the Shannon diversity was significantly increased in the SS group compared to the control group, particularly in a non-dry condition (Fig 1B), and it showed a positive correlation with bacterial load (Fig 1C). Weighted UniFrac-based PCoA and PERMANOVA tests revealed that bacterial communities were significantly different by the disease ($P = 0.001$) in both non-dry ($P = 0.048$) and dry

**Table 1. Characteristics of the subjects used for oral microbiota analysis and sequencing data.**

| | Subgroup | Control (n = 25) | SS (n = 25) | *P* value |
|---|---|---|---|---|
| Oral dryness[a], *n* (%) | | 11 (44) | 17 (68) | 0.154[c] |
| Salivary gland hypofunction[b], *n* (%) | | 14 (56) | 17 (68) | 0.561[c] |
| Age, mean ± SD | All | 50.4 ± 17.1 | 53.2 ± 10.6 | 0.484[d] |
| | Without dryness | 39.6 ± 10.9 | 51.4 ± 10.7 | 0.141[e] |
| | Dryness | 64.2 ± 13.2 | 54.1 ± 10.8 | 0.160[e] |
| UWSFR, median (range) | All | 0.24 (0.02–1.56) | 0.10 (0–0.50) | 0.125[f] |
| | Without dryness | 0.37 (0.23–1.56) | 0.30 (0.25–0.50) | 1.000[g] |
| | Dryness | 0.05 (0.02–0.10) | 0.10 (0–0.10) | 1.000[g] |
| Valid read count, mean ± SD | All | 10246 ± 2363 | 8706 ± 2213 | 0.021[d] |
| | Without dryness | 10718 ± 2217 | 7982 ± 937 | 0.056[e] |
| | Dryness | 9645 ± 2511 | 9047 ± 2565 | 1.000[e] |
| Read length, mean ± SD | All | 461 ± 2.9 | 443 ± 4.1 | < 0.0005[d] |
| | Without dryness | 461 ± 1.9 | 445 ± 2.3 | < 0.0005[e] |
| | Dryness | 461 ± 3.9 | 442 ± 4.5 | < 0.0005[e] |
| Goods Lib. Coverage, mean ± SD | All | 0.992±0.003 | 0.991±0.003 | 0.176[d] |
| | Without dryness | 0.992±0.003 | 0.989±0.003 | 0.047[e] |
| | Dryness | 0.992±0.003 | 0.992±0.002 | 0.878[e] |

SS: Sjögren's syndrome; UWSFR: unstimulated whole salivary flow rate

[a], UWSFR ≤ 0.1 ml/min;

[b], UWSFR < 0.25 ml/min;

[c], by Fisher's exact test;

[d], by t-test;

[e], by ANOVA with Bonferroni post hoc test;

[f], by Mann-Whitney test;

[g], by Kruskal-Wallis with Bonferroni post hoc test

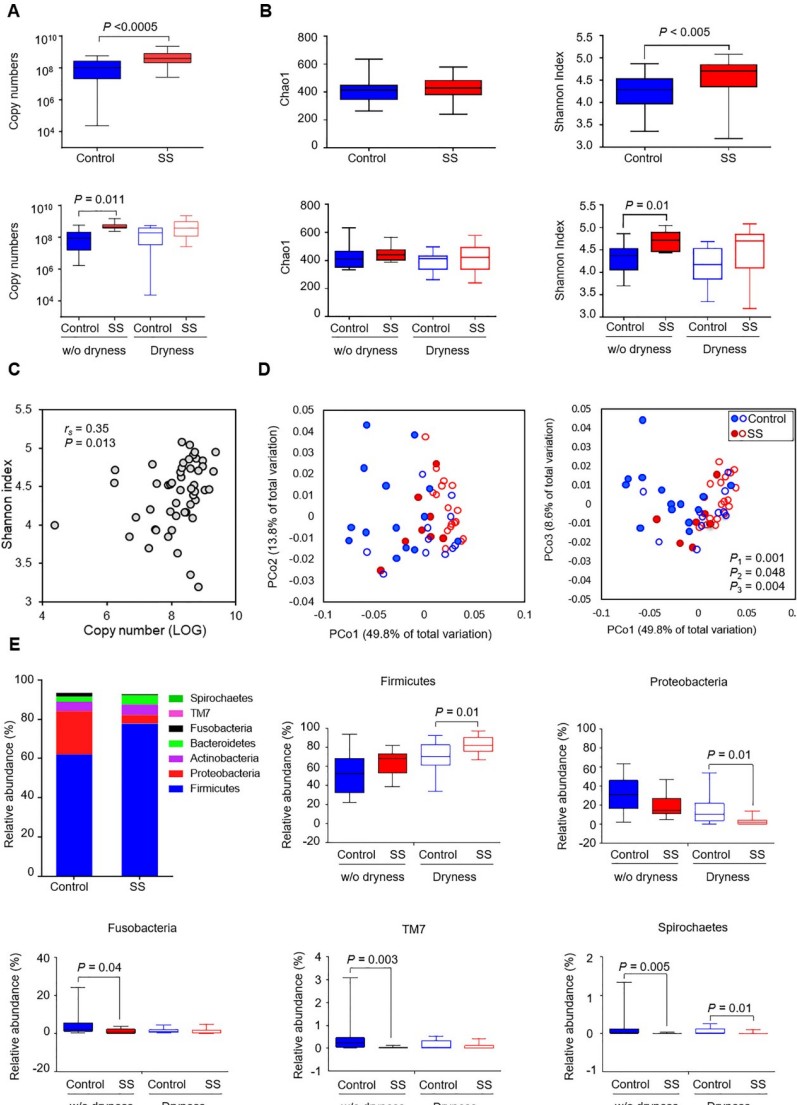

**Fig 1. Dysbiosis of oral microbiota in patients with Sjögren syndrome (SS).** Oral bacterial communities collected by whole mouthwash from control, including healthy subjects ($n$ = 15) and drug-induced dry mouth patients ($n$ = 10), and SS patients ($n$ = 25) were analyzed by pyrosequencing of the 16S rRNA gene. The control and SS groups were divided into subgroups, control without dryness ($n$ = 14), control with dryness ($n$ = 11), SS without dryness ($n$ = 8), and SS with dryness ($n$ = 17). (A) The total bacterial load in each sample was estimated by real-time PCR using universal primers targeting the bacterial 16S rRNA gene. The total bacterial load was expressed as the 16S rRNA gene copy number in the total DNA obtained from each sample. (B) Measures of alpha diversity (Chao1 and Shannon indices) are presented. (C) Correlation plot between the total bacterial load and the Shannon index. (D) PCoA plot generated using a weighted UniFrac metric. Empty symbols indicate subjects with dryness. $R^2$ and $P$ values by PERMANOVA (adonis) test. $P_1$: control ($n$ = 25) vs. SS ($n$ = 25), $P_2$: control without dryness vs. SS without dryness, $P_3$: control with dryness vs. SS with dryness. (E) The mean relative abundance levels of the top 7 phyla in control ($n$ = 25) and SS ($n$ = 25) are presented as a column graph. *, $P$ < 0.05 by Mann-Whitney U test. For the 5 phyla with significant intergroup differences, the relative abundance of control and SS subgroups, i.e., either without (w/o) dryness or with dryness, are presented as boxplots. $P$ values by Kruskal-Wallis test followed by post hoc analysis with Bonferroni adjustment.

($P$ = 0.004) conditions (Fig 1D). Comparing the taxonomic compositions revealed that the oral microbiota in SS was characterized by increased Firmicutes relative abundance but decreases in the relative abundances of Proteobacteria, Fusobacteria, TM7, and Spirochaetes (Fig 1E). At the genus level, 5 genera, including *Streptococcus*, *Prevotella*, *Lactobacillus*,

*Atopobium*, and *Staphylococcus*, were enriched but 34 genera were underrepresented in SS compared to control, among which 9 and 8 genera maintained significant differences in non-dry and dry conditions, respectively (S2 Table).

To find SS-associated species, LEfSe and logistic regression analysis were applied to compare control vs. SS in combined samples, non-dry, or dry conditions. Among the 63 species discriminating SS from control obtained by the LEfSe analysis, 17 species overlapped with 31 species that discriminate SS in the non-dry condition, and 25 species overlapped with 36 species that discriminate SS in the dry condition (Tables 2–4). When the species that had significant negative correlations with UWSFR in the control group were excluded, several *Prevotella* and *Veillonella* species were associated with SS in non-dry and dry conditions, respectively. Logistic regression analysis revealed only 1 to 3 species with significant associations for each outcome (Table 5). Collectively, these results identify SS-specific changes in oral microbiota, independent of oral dryness.

## Deregulation of SGECs by *P. melaninogenica in vitro*

To investigate potential roles of the SS-associated species in the pathogenesis of SS, we tested *in vitro* if selected species of oral bacteria could induce functional and phenotypic changes in SGECs. Three SS-associated bacterial species were selected among the species that are top 30 abundant and contain aquaporins (AQPs) or porins homologous to human AQP 5 [25]: *P. histicola*, a species associated with SS in both non-dry and dry conditions; and *P. melaninogenica* and *R. mucilaginosa*, species associated with SS in the non-dry condition. *S. salivarius*, the most abundant commensal species in saliva, and *F. nucleatum*, a highly immune stimulatory species, were additionally included as negative and positive controls, respectively.

To determine if exposure to bacteria can induce APC-like phenotypes in SGECs, we challenged HSG cells with the selected bacterial species and examined the expression levels of MHC I, MHC II and costimulatory molecules. The baseline expression levels of MHC II, CD80, and CD86 in HSG cells were barely detectable, as assessed by fluorescence microscopy and/or flow cytometry. However, stimulation with IFNγ upregulated expression of these molecules as well as that of MHC I (S2 Fig). Exposure to *F. nucleatum* and *P. melaninogenica* was sufficient to upregulate MHC I expression in the absence of IFNγ, while *R. mucilaginosa* downregulated MHC I and CD86 under the same condition (Fig 2A). In the presence of IFNγ, *F. nucleatum* and *P. melaninogenica* further upregulated the expression levels of MHC I, MHC II, and CD80, but *S. salivarius* and *R. mucilaginosa* downregulated the expression levels of MHC I and CD86 (Fig 2B). The upregulation of MHC molecules may be associated with the ability of bacteria to invade host cells. Bacterial invasion of HSG cells was determined using bacteria double-stained with CFSE and pHrodoRed dyes. The pHrodoRed dye is non-fluorescent at neutral pH but becomes fluorescent at acidic pH. Confocal microscopy showed a sizable amount of double-fluorescence, i.e., intracellular endosomal location, by *R. mucilaginosa*, *F. nucleatum*, and *P. melaninogenica*, while the other species presented much less (Fig 2C).

Intracellular bacteria can induce IFNλ production in epithelial cells [26]. Thus, we assessed the effect of bacterial infection on IFNλ production by HSG cells. *F. nucleatum* and *P. melaninogenica* increased IFNλ secretion, while *R. mucilaginosa* and *P. histicola* decreased it (Fig 2D). Altogether, among the selected SS-associated species, only *P. melaninogenica* induced the deregulation of HSG cells.

## The presence of bacteria within ductal cells and the infiltration areas of LSGs

We next asked whether bacteria can reach the salivary glands to induce deregulation of SGECs. The sections of LSG biopsies from 8 non-SS and 15 SS patients were obtained, and

**Table 2. Species associated with control (n = 25) vs. SS (n = 25).**

| Name | Relative abundance[a] | | LEfSe | | Spearman's rho |
|---|---|---|---|---|---|
| | Control | SS | LDA Score | P | |
| *Streptococcus*_HQ748137_s | 0.83 (4.16) | 2.32 (19.49) | 4.53 | 0.012 | - 0.44* |
| *Prevotella histicola* | 0.06 (4.48) | 0.86 (10.46) | 4.33 | 0.016 | 0.27 |
| *Streptococcus*_HQ762034_s | 0.63 (9.11) | 1.64 (8.20) | 4.31 | 0.026 | - 0.28 |
| *Streptococcus parasanguinis* | 1.11 (4.94) | 1.92 (3.56) | 4.28 | 0.028 | - 0.49* |
| *Streptococcus*_4P003152_s | 0.26 (4.11) | 0.80 (4.24) | 4.08 | 0.004 | - 0.21 |
| *Streptococcus_uc* | 0.68 (3.45) | 1.15 (1.49) | 4.07 | 0.012 | - 0.41* |
| *Actinomyces*_4P002811_s | 0.07 (0.90) | 0.33 (3.27 | 3.77 | 0.012 | - 0.35 |
| *Streptococcus anginosus* | 0.02 (1.44) | 0.11 (8.08) | 3.72 | 0.082 | - 0.19 |
| *Streptococcus mutans* | 0.00 (0.19) | 0.03 (7.14) | 3.56 | 0.026 | - 0.19 |
| *Haemophilus*_HQ807753_s | 0.01 (0.10) | 0.08 (2.81) | 3.46 | 0.044 | 0.39 |
| *Veillonella parvula* | 0.01 (0.17) | 0.08 (3.20) | 3.45 | 0.020 | - 0.17 |
| *Prevotella*_FM995711_s | 0.00 (3.28) | 0.02 (2.25) | 3.44 | 0.016 | 0.11 |
| *Streptococcus sobrinus* | 0.00 (0.03) | 0.00 (4.55) | 3.42 | 0.048 | - 0.28 |
| *Atopobium parvulum* | 0.03 (0.90) | 0.12 (1.21) | 3.37 | 0.064 | - 0.05 |
| *Prevotella salivae* | 0.01 (1.00) | 0.11 (1.15) | 3.33 | 0.032 | 0.24 |
| *Lactobacillus salivarius* | 0.00 (0.06) | 0.00 (2.44) | 3.32 | 0.030 | 0.15 |
| *Veillonella rodentium* | 0.01 (0.22) | 0.06 (1.28) | 3.21 | 0.010 | 0.10 |
| *Streptococcus intermedius* | 0.01 (0.13) | 0.04 (1.36) | 3.18 | 0.078 | - 0.08 |
| *Haemophilus haemolyticus* | 0.00 (0.25) | 0.00 (0.78) | 2.85 | 0.040 | 0.33 |
| *Lactobacillus fermentum* | 0.00 (0.00) | 0.00 (0.69) | 2.79 | 0.002 | NA |
| *Prevotella*_4P003758_s | 0.00 (0.18) | 0.02 (0.52) | 2.70 | 0.008 | - 0.23 |
| *Prevotella*_4P005351_s | 0.00 (0.38) | 0.01 (0.29) | 2.67 | 0.040 | 0.15 |
| *Treponema denticola* | 0.00 (0.64) | 0.00 (0.08) | 2.51 | 0.050 | - 0.01 |
| *Saccharimonas*_AM420132_s | 0.00 (0.43) | 0.00 (0.05) | 2.53 | 0.010 | 0.22 |
| *Leptotrichia*_FJ976402_s | 0.01 (0.21) | 0.00 (0.05) | 2.53 | 0.002 | 0.11 |
| DQ241813_g_DQ241813_s | 0.02 (0.21) | 0.00 (0.09) | 2.55 | 0.028 | 0.09 |
| *Eikenella corrodens* | 0.02 (0.19) | 0.00 (0.09) | 2.56 | 0.002 | 0.29 |
| *Neisseria*_AY005028_s | 0.00 (0.35) | 0.00 (0.03) | 2.60 | 0.030 | - 0.16 |
| *Campylobacter showae* | 0.01 (0.31) | 0.00 (0.04) | 2.62 | 0.000 | 0.17 |
| *Leptotrichia hofstadii* | 0.01 (0.24) | 0.00 (0.06) | 2.65 | 0.018 | 0.23 |
| *Prevotella shahii* | 0.00 (0.30) | 0.00 (0.08) | 2.66 | 0.028 | 0.20 |
| AM420062_g_AM420062_s | 0.00 (0.45) | 0.00 (0.20) | 2.79 | 0.082 | 0.12 |
| *Prevotella nigrescens* | 0.01 (0.73) | 0.00 (0.07) | 2.80 | 0.000 | 0.73** |
| *Stomatobaculum longum* | 0.03 (0.33) | 0.00 (0.27) | 2.85 | 0.020 | 0.22 |
| AF432141_g_AF432141_s | 0.00 (1.32) | 0.00 (0.00) | 2.87 | 0.004 | 0.28 |
| *Capnocytophaga gingivalis* | 0.04 (0.48) | 0.00 (0.04) | 2.88 | 0.000 | - 0.21 |
| *Leptotrichia wadei* | 0.02 (1.02) | 0.00 (0.11) | 2.91 | 0.002 | 0.09 |
| *Capnocytophaga leadbetteri* | 0.00 (0.87) | 0.00 (0.11) | 2.92 | 0.034 | 0.08 |
| *Leptotrichia hongkongensis* | 0.03 (1.24) | 0.00 (0.27) | 2.96 | 0.048 | - 0.04 |
| *Neisseria_uc* | 0.05 (0.53) | 0.00 (0.41) | 2.98 | 0.002 | 0.20 |
| *Oribacterium asaccharolyticum* | 0.04 (1.08) | 0.00 (0.33) | 3.03 | 0.024 | 0.30 |
| *Halomonas hamiltonii* | 0.00 (1.99) | 0.00 (0.00) | 3.05 | 0.020 | 0.08 |
| *Saccharimonas*_AF385520_s | 0.03 (1.06) | 0.00 (0.34) | 3.11 | 0.020 | 0.27 |
| *Corynebacterium matruchotii* | 0.07 (1.47) | 0.00 (0.57) | 3.12 | 0.016 | 0.29 |
| *Lachnoanaerobaculum orale* | 0.09 (0.60) | 0.04 (0.24) | 3.16 | 0.026 | - 0.13 |
| *Streptococcus*_HQ757980_s | 0.04 (2.00) | 0.13 (0.37) | 3.19 | 0.020 | - 0.19 |

(*Continued*)

**Table 2.** (Continued)

| Name | Relative abundance[a] | | LEfSe | | Spearman's rho |
|---|---|---|---|---|---|
| | Control | SS | LDA Score | P | |
| *Veillonella rogosae* | 0.09 (0.69) | 0.01 (0.22) | 3.19 | 0.002 | 0.58** |
| ***Leptotrichia_DQ447842_s*** | 0.06 (1.55) | 0.00 (0.70) | 3.32 | 0.022 | 0.36 |
| *Porphyromonas gingivalis* | 0.01 (2.22) | 0.00 (0.17) | 3.38 | 0.034 | - 0.21 |
| *Campylobacter concisus* | 0.07 (2.73) | 0.00 (0.21) | 3.42 | 0.002 | 0.43* |
| ***Porphyromonas_AM420091_s*** | 0.08 (3.51) | 0.00 (2.28) | 3.53 | 0.030 | 0.06 |
| *Neisseria oralis* | 0.01 (4.81) | 0.00 (0.31) | 3.54 | 0.046 | 0.18 |
| *Haemophilus sputorum* | 0.07 (4.44) | 0.00 (0.15) | 3.57 | 0.000 | 0.44* |
| *Neisseria elongata* | 0.14 (2.36) | 0.00 (2.40) | 3.60 | 0.004 | 0.36 |
| *Neisseria sicca group* | 0.07 (6.68) | 0.00 (0.64) | 3.72 | 0.034 | 0.06 |
| *Rothia aeria* | 0.29 (3.44) | 0.04 (1.09) | 3.74 | 0.016 | 0.02 |
| *Lautropia mirabilis* | 0.10 (6.46) | 0.00 (0.24) | 3.82 | 0.000 | 0.44* |
| *Fusobacterium periodonticum* | 0.68 (2.72) | 0.03 (3.67) | 3.94 | 0.008 | 0.16 |
| *Streptococcus sanguinis* | 1.03 (5.92) | 0.35 (2.85) | 4.13 | 0.012 | 0.30 |
| ***Streptococcus_CP006776_s*** | 1.96 (16.7) | 0.59 (6.65) | 4.53 | 0.022 | 0.06 |
| *Neisseria perflava* | 0.79 (50.2) | 0.00 (11.03) | 4.81 | 0.004 | 0.19 |
| *Neisseria subflava* | 3.71 (33.7) | 0.02 (31.99) | 4.82 | 0.002 | 0.35 |
| *Haemophilus parainfluenzae* | 3.54 (32.7) | 1.22 (9.75) | 4.84 | 0.022 | 0.42* |

[a]Expressed as the median (range)

*, $P < 0.05$;

**, $P < 0.01$

ISH was performed using universal and *P. melaninogenica*-specific probes targeting the 16S rRNA gene. Because the non-SS samples were obtained from subjects who also had signs of SS, 7 samples had nonspecific chronic inflammation, FLS with scores < 1, or sclerosing chronic sialadenitis. Among the 15 SS samples, 13 samples had FLS with scores ≥ 1, and the others had FLS with scores < 1 or nonspecific chronic inflammation. In the LSGs without inflammation, only a few bacteria were detected within ductal cells. However, in the LSGs with inflammation, the ductal cells were heavily infected with bacteria, and the infection spread to the area of inflammatory infiltration and adjacent acini (Fig 3A). Importantly, the ducts in the area without inflammation were also infected with bacteria in the LSGs with FLS from SS patients (S3 Fig). At the FLS foci with GC-like structures, bacterial infection was observed at the GC-like structure as well as at the ducts (Fig 3B). Compared to the universal probe, the *P. melaninogenica*-specific probe detected a much lower number of bacteria (Fig 3A and 3B). When the signal intensities were measured, the SS group tended to have higher levels of bacteria than the non-SS group by both the universal and *P. melaninogenica*-specific probes without statistical significance (Fig 3C and 3D).

## Discussion

Here, we propose that dysbiotic oral microbiota may cause epithelitis and FLS, which is based on the results of oral microbiota analysis, deregulation of SGECs *in vitro* by one of the SS-associated species *P. melaninogenica*, and the presence of bacteria, including *P. melaninogenica*, within the ductal cells and the area of infiltration in the LSG biopsies.

Different anatomical sites in the oral cavity, such as tooth, gingival sulcus, tongue, cheek, and palate, provide highly heterogeneous ecological environments and are colonized by

**Table 3. Species associated with SS normalcy (n = 8) vs. control normalcy (n = 14).**

| | Relative abundance[a] | | LeFSe | | Spearman's rho |
| --- | --- | --- | --- | --- | --- |
| | Control w/o dryness | SS w/o dryness | LDA Score | P | |
| Rothia mucilaginosa | 0.67 (4.39) | 2.67 (10.01) | 4.57 | 0.048 | -0.442* |
| Granulicatella adiacens | 0.95 (2.90) | 3.31 (5.66) | 4.47 | 0.058 | -0.408* |
| Prevotella melaninogenica | 0.31 (3.40) | 2.37 (4.50) | 4.41 | 0.016 | 0.270 |
| Prevotella histicola | 0.08 (4.48) | 0.69 (10.45) | 4.38 | 0.096 | 0.271 |
| Streptococcus_HQ762034_s | 0.29 (2.30) | 1.46 (8.20) | 4.32 | 0.028 | -0.280 |
| Streptococcus_HQ748137_s | 0.60 (2.75) | 2.08 (3.93) | 4.30 | 0.058 | -0.443* |
| Streptococcus parasanguinis | 0.56 (1.60) | 1.93 (2.26) | 4.25 | 0.012 | -0.485* |
| Streptococcus_uc | 0.35 (1.20) | 1.05 (0.69) | 4.03 | 0.010 | -0.413* |
| Streptococcus_AFQU_s | 0.05 (0.55) | 0.20 (2.06) | 3.62 | 0.068 | 0.037 |
| Actinomyces_4P002811_s | 0.05 (0.50) | 0.32 (1.36) | 3.62 | 0.068 | -0.348 |
| Streptococcus_JF120814_s | 0.11 (0.66) | 0.33 (0.73) | 3.60 | 0.068 | 0.050 |
| Prevotella_FM995711_s | 0.00 (0.10) | 0.21 (1.27) | 3.55 | 0.004 | 0.107 |
| Haemophilus_HQ807753_s | 0.02 (0.10) | 0.24 (0.78) | 3.50 | 0.000 | 0.392 |
| Prevotella_FJ557895_s | 0.01 (0.38) | 0.08 (0.81) | 3.40 | 0.084 | 0.267 |
| Prevotella salivae | 0.02 (1.00) | 0.12 (0.43) | 3.21 | 0.048 | 0.235 |
| Streptococcus_HQ757980_s | 0.02 (0.17) | 0.07 (0.30) | 3.08 | 0.040 | -0.193 |
| Streptococcus_GU374045_s | 0.00 (0.13) | 0.03 (0.25) | 2.79 | 0.010 | -0.114 |
| Streptococcus sobrinus | 0.00 (0.00) | 0.01 (0.29) | 2.73 | 0.010 | -0.281 |
| Prevotella_4P003758_s | 0.00 (0.04) | 0.04 (0.11) | 2.66 | 0.006 | -0.229 |
| Selenomonas_g1_4P003112_s | 0.01 (0.16) | 0.00 (0.01) | 2.51 | 0.078 | 0.431* |
| Leptotrichia_FJ976402_s | 0.03 (0.08) | 0.00 (0.05) | 2.53 | 0.056 | 0.110 |
| Treponema socranskii | 0.00 (0.37) | 0.00 (0.00) | 2.58 | 0.074 | 0.332 |
| Prevotella_FJ577255_s | 0.02 (0.26) | 0.00 (0.06) | 2.67 | 0.040 | 0.280 |
| Alloprevotella tannerae | 0.01 (0.32) | 0.00 (0.00) | 2.69 | 0.022 | 0.217 |
| Campylobacter gracilis | 0.02 (0.37) | 0.00 (0.03) | 2.73 | 0.064 | 0.456* |
| Campylobacter showae | 0.02 (0.31) | 0.00 (0.02) | 2.76 | 0.022 | 0.171 |
| Prevotella nigrescens | 0.06 (0.73) | 0.00 (0.00) | 3.03 | 0.002 | 0.725** |
| Alloprevotella_FM996479_s | 0.03 (1.11) | 0.00 (0.05) | 3.09 | 0.066 | 0.500* |
| AF432141_g_AF432141_s | 0.00 (1.32) | 0.00 (0.00) | 3.11 | 0.074 | 0.275 |
| Saccharimonas_AF385520_s | 0.08 (1.06) | 0.00 (0.08) | 3.24 | 0.016 | 0.274 |
| Oribacterium sinus | 0.08 (0.72) | 0.03 (0.12) | 3.25 | 0.034 | 0.213 |
| Veillonella rogosae | 0.17 (0.67) | 0.03 (0.22) | 3.38 | 0.034 | 0.580** |
| Campylobacter concisus | 0.12 (2.72) | 0.02 (0.09) | 3.62 | 0.006 | 0.428* |
| Veillonella tobetsuensis | 0.56 (1.34) | 0.07 (0.45) | 3.74 | 0.068 | 0.463* |
| Haemophilus sputorum | 0.21 (4.44) | 0.02 (0.15) | 3.80 | 0.032 | 0.444* |
| Lautropia mirabilis | 0.18 (6.46) | 0.03 (0.21) | 3.99 | 0.040 | 0.436* |
| Haemophilus parainfluenzae | 6.78 (32.37) | 2.86 (9.30) | 5.02 | 0.096 | 0.419* |

[a]Expressed as the median (range)

*, P < 0.05;

**, P < 0.01

unique bacterial communities [27]. We had an idea that oral bacteria may reach and affect the pathophysiology of minor salivary glands distributed through labial, buccal, palatal, and lingual mucosae. Thus, we chose to collect oral bacteria by whole mouthwash. While previous studies reported no difference or a decreased Shannon diversity in SS [10, 11, 13, 14, 17], we

**Table 4. Species associated with SS dryness (n = 17) vs. control dryness (n = 11).**

| | Relative abundance[a] | | LeFSe | | Spearman's rho |
|---|---|---|---|---|---|
| | **Control with dryness** | **SS with dryness** | **LDA Score** | **P** | |
| *Veillonella dispar* | 0.78 (12.98) | 2.58 (8.43) | 4.50 | 0.082 | 0.380 |
| *Prevotella histicola* | 0.03 (3.78) | 0.97 (7.36) | 4.30 | 0.080 | 0.271 |
| *Streptococcus_4P003152_s* | 0.60 (4.10) | 1.22 (4.17) | 4.17 | 0.092 | -0.212 |
| *Veillonella parvula* | 0.01 (0.17) | 0.09 (3.20) | 3.59 | 0.046 | -0.166 |
| *Lactobacillus salivarius* | 0.00 (0.02) | 0.01 (2.44) | 3.48 | 0.034 | 0.146 |
| *Veillonella tobetsuensis* | 0.04 (0.58) | 0.15 (1.13) | 3.43 | 0.048 | 0.463* |
| *Veillonella rodentium* | 0.01 (0.06) | 0.12 (1.28) | 3.34 | 0.002 | 0.101 |
| *Lactobacillus fermentum* | 0.00 (0.00) | 0.00 (0.69) | 2.95 | 0.018 | |
| *Rothia_uc* | 0.01 (0.11) | 0.00 (0.14) | 2.51 | 0.078 | -0.347 |
| *Leptotrichia_FJ976402_s* | 0.00 (0.21) | 0.00 (0.04) | 2.54 | 0.076 | 0.110 |
| *Haemophilus sputorum* | 0.00 (0.19) | 0.00 (0.00) | 2.62 | 0.006 | 0.444* |
| *Neisseria_AY005028_s* | 0.01 (0.24) | 0.00 (0.02) | 2.63 | 0.012 | -0.161 |
| *Neisseria_uc* | 0.04 (0.14) | 0.00 (0.03) | 2.70 | 0.000 | 0.198 |
| *Neisseria oralis* | 0.00 (0.69) | 0.00 (0.05) | 2.87 | 0.088 | 0.180 |
| *Capnocytophaga gingivalis* | 0.02 (0.48) | 0.00 (0.00) | 2.89 | 0.000 | 0.096 |
| *Capnocytophaga leadbetteri* | 0.00 (0.87) | 0.00 (0.02) | 2.97 | 0.076 | 0.078 |
| *Leptotrichia wadei* | 0.02 (1.02) | 0.00 (0.05) | 3.07 | 0.036 | 0.087 |
| *Porphyromonas gingivalis* | 0.06 (0.53) | 0.00 (0.15) | 3.07 | 0.006 | -0.212 |
| *Porphyromonas_AM420091_s* | 0.06 (0.93) | 0.00 (1.77) | 3.26 | 0.014 | 0.060 |
| *Lachnoanaerobaculum orale* | 0.11 (0.60) | 0.03 (0.24) | 3.27 | 0.048 | -0.132 |
| *Lautropia mirabilis* | 0.01 (2.31) | 0.00 (0.24) | 3.44 | 0.004 | 0.436* |
| *Neisseria elongata* | 0.03 (2.15) | 0.00 (0.02) | 3.45 | 0.004 | 0.362 |
| *Fusobacterium periodonticum* | 0.51 (1.59) | 0.03 (3.67) | 3.75 | 0.094 | 0.161 |
| *Rothia aeria* | 0.29 (3.44) | 0.01 (0.52) | 3.86 | 0.016 | 0.016 |
| *Neisseria sicca group* | 0.07 (6.68) | 0.00 (0.46) | 3.91 | 0.046 | 0.060 |
| *Neisseria mucosa* | 0.00 (7.60) | 0.00 (1.88) | 4.10 | 0.034 | 0.094 |
| *Streptococcus sanguinis* | 0.51 (4.78) | 0.26 (2.85) | 4.14 | 0.082 | 0.303 |
| *Neisseria subflava* | 1.43 (11.65) | 0.00 (0.76) | 4.40 | 0.018 | 0.350 |
| *Streptococcus_CP006776_s* | 1.96 (16.72) | 0.49 (2.38) | 4.59 | 0.040 | 0.057 |
| *Neisseria perflava* | 0.69 (50.17) | 0.00 (4.29) | 4.91 | 0.016 | 0.192 |

[a]Expressed as the median (range)

*, $P < 0.05$;

**, $P < 0.01$

**Table 5. Species associated with SS in combined, non-dry, and dry conditions identified by logistic regression analysis.**

| | **Species** | **Odd Ratio** | **Confidence Interval** | **P value** |
|---|---|---|---|---|
| control vs. SS | *Prevotella melaninogenica* | 22.4 | 2.5–202.5 | 0.006 |
| | *Veillonella rogosae* | 0.006 | 0.005–0.009 | 0.007 |
| | *Eikenella corrodens* | 0.006 | 0.004–0.01 | 0.034 |
| Control w/o dryness vs. SS w/o dryness | *Haemophilus_HQ807753_s* | 1.3E+16 | 27–7E+30 | 0.031 |
| control with dryness vs. SS with dryness | *Neisseria_uc* | 0.006 | 0.004–0.01 | 0.008 |

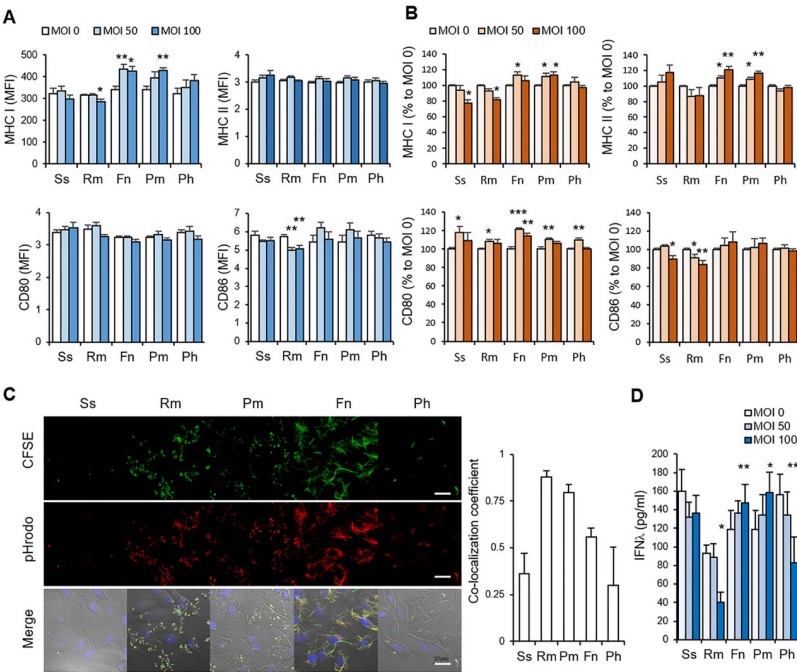

**Fig 2. Deregulation of SGECs by *P. melaninogenica in vitro*.** (A and B) HSG cells were cocultured with *S. salivarius* (Ss), *R. mucilaginosa* (Rm), *F. nucleatum* (Fn), *P. melaninogenica* (Pm), and *P. histicola* (Ph) at MOIs of 0, 50, and 100 for 3 days in the absence (A) or presence (B) of IFNγ. The expression levels of MHC I, MHC II, CD80, and CD86 were analyzed by flow cytometry. The data are presented as the means ± SEMs of 3 experiments performed in duplicate. The data obtained in the presence of IFNγ were normalized to noninfected control cells (MOI 0) to normalize the variable effects of IFNγ from experiment to experiment. *, $P < 0.05$; ** $P < 0.01$ versus MOI 0 by t-test. (C) HSG cells grown on coverslips were infected with each bacterial species, which had been double-stained with CFSE and pHrodo Red, at an MOI of 100 for 6 hours. The cells were examined by confocal microscopy. The scale bars indicate 20 μm. The colocalization coefficient of pHrodo over CFSE is shown in the graph in the right panel. (D) HSG cells were cocultured with each oral bacterial species at MOIs of 0, 50, and 100 for 3 days in the absence of IFNγ. The concentrations of IFN-λ1/3 in the culture supernatant were measured by ELISA. The data are presented as the means ± SEMs of 3 experiments performed in duplicate. *, $P < 0.05$; ** $P < 0.01$ versus MOI 0 by t-test.

observed a significantly increased Shannon diversity in SS compared to control, particularly in the non-dry condition. The salivary microbiota of individuals with dental caries is characterized with a reduced Shannon diversity, which may be attributed to decreased salivary pH [28]. Thus, the decreased Shannon diversity in the previous studies may reflect increased caries prevalence in SS. Although the oral health status of participating subjects was not evaluated in this study, the caries prevalence may not be different between the control and SS subgroups without oral dryness. Interestingly, the Shannon index had a weak but significant positive correlation with total bacterial load that was also significantly increased in SS compared to control. Decreased expression of human beta-defensins in the salivary glands of SS patients [29] may contribute to the increased bacterial load in the oral cavity.

Considering the differences in the sampling sites (whole mouthwash, saliva, tongue, and buccal mucosa), ethnicity, diet, and the sequenced V regions of 16S rRNA gene, the results of oral microbiota analysis cannot be directly compared with those of previous studies. Nevertheless, there are several common findings. Increased relative abundances of phylum Firmicutes [10, 13] and genera *Streptococcus* [11–13], *Lactobacillus* [10–12], *Prevotella* [11], and *Atopobium* [10] but decreased relative abundances of phyla Proteobacteria [10, 12] and Spirochaetes [13] and genera *Haemophilus* [10–12], *Leptotrichia* [11], *Fusobacterium* [11], *Lautropia* [10,

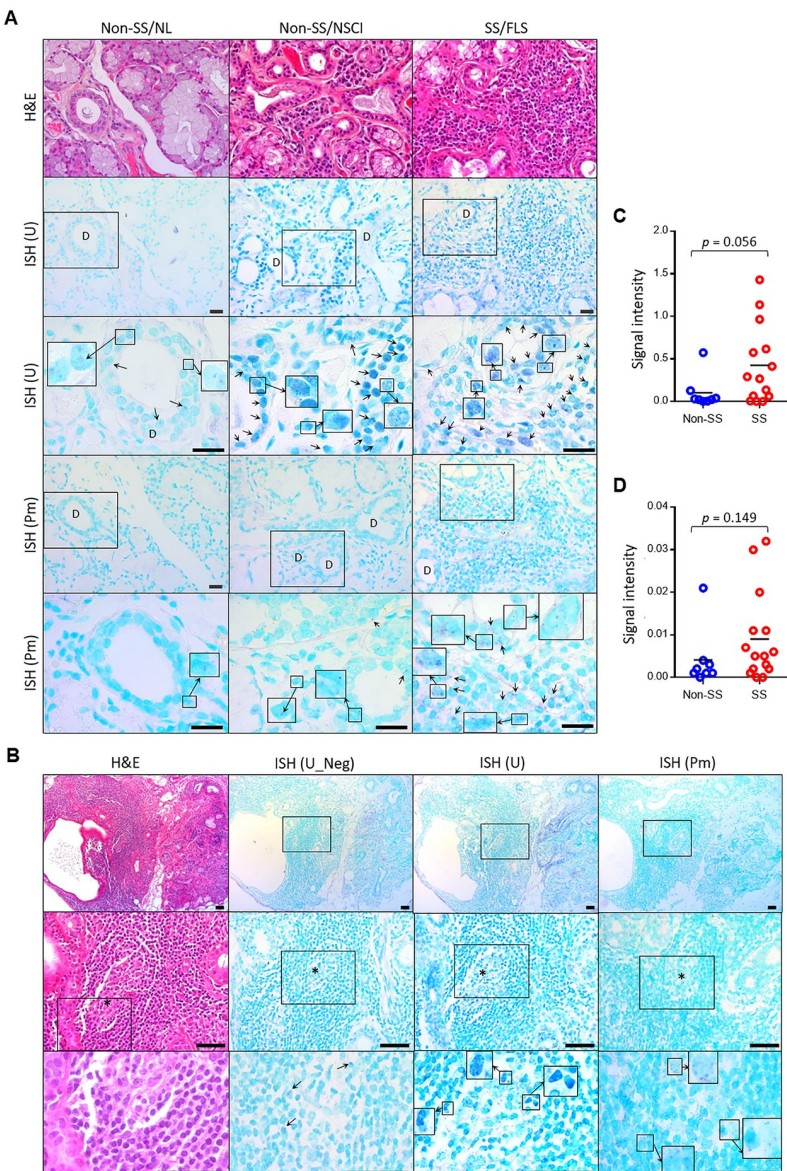

**Fig 3. The presence of bacteria, including *P. melaninogenica*, within ductal cells and the infiltration areas of LSGs.**
The sections of paraffin-embedded LSG tissues obtained from non-SS subjects or SS patients were subjected to H&E staining and *in situ* hybridization (ISH) using universal (U) or *P. melaninogenica*-specific (Pm) probes. As the negative control (Neg), sections were hybridized with a probe mixed with a 10-fold amount of unlabeled probe. (A) Representative areas of normal (NL) histology, nonspecific chronic inflammation (NSCI), and focal lymphocytic sialadenitis (FLS) are shown. Selected areas (square) were examined with higher magnification. Arrows indicate ISH signals in violet color and the shape of rod or cocci. D, duct. Scale bars indicate 25 μm. (B) A represented area of FLS with a germinal center (GC)-like structure is shown. The light zone is marked with *. Arrows indicate ISH signals in violet color with the shapes of rods or cocci. Several strong ISH signals were not completely inhibited in the negative control. Scale bars indicate 50 μm. (C and D) The signal intensities of ISH performed using the universal (C) or Pm-specific (D) probes were measured.

11], and *Neisseria* [10–12] in SS compared to those in controls have been repeatedly observed, including within the current study. The reduced relative abundance of *P. gingivalis* also agrees with previous studies [11, 13] and explains the lack of increased risk of periodontitis in SS patients, despite the reduced salivary antimicrobial function [30, 31].

To find SS-specific changes independent of oral dryness, we compared the control and SS groups after stratifying each group into without or with dryness subgroups. We also considered the correlations between each taxon and UWSFR observed in the control subjects. The relative abundances of *P. melaninogenica*, *V. dispar*, and *P. histicola* were increased in the non-dry, dry, and both conditions of SS, respectively, but had a tendency to decrease with reduced UWSFR (positive Spearman's rho value). Likewise, *P. gingivalis* relative abundance was decreased in SS, despite the negative rho value. Thus, these are more likely to be true SS-specific changes. In contrast, *R mucilaginosa*, the species associated with SS in the non-dry condition, had a significant negative correlation with UWSFR, suggesting the potential confounding effect of reduced salivary rate. The relative abundances of cariogenic species *S. mutans* and *S. sobrinus* were increased in SS compared to control when all subjects were included, but not in the subgroups with dryness, suggesting the confounding effect of reduced salivation and the aciduric capacity of bacteria.

It is noteworthy that antimicrobial activities have been observed by a number of non-antibiotic drugs such as the beta-adrenergic receptor antagonists, diuretic drugs, H1 antihistamines, psychotropic drugs, non-steroid anti-inflammatory drugs, and proton pump inhibitors [32, 33], and some of the medications taken by the drug-related sicca patients are known to have antibacterial activity (S2 Table). Although the control subjects without or with dryness harbored comparable levels of total bacterial loads, the medications with antimicrobial activity may have affected the composition of oral microbiota.

The five selected oral bacterial species variably regulated the expression of IFNλ and proteins involved in antigen presentation. All selected species belong to normal human oral flora but can become pathogenic in immunocompromised hosts or ectopic sites. Interestingly, Gram-negative *P. melaninogenica* and *F. nucleatum* upregulated expression of MHC molecules and CD80, while Gram-positive species downregulated them. *P. melaninogenica* and *F. nucleatum* also upregulated IFNλ production. IFNλ is classified as a type III IFN that shares induction stimuli and functional similarities with type I IFN [34]. Epithelial cells exclusively express IFNλ [35]. Although only viral infection has been suspected as the cause of the IFN signature in SS, among the pattern recognition receptors mediating type I and III IFN induction [36], TLR4 and TLR9 can recognize bacterial components. Gram-negative bacteria that invade host cells, such as *F. nucleatum* and *P. melaninogenica*, would provide ligands for both TLR4 and TLR9, leading to the induction of type I & III IFN. Since cell-mediated immunity is important for protection against intracellular pathogens, the upregulation of MHC expression by bacteria that invade HSG cells is reasonable. Why *R. mucilaginosa* suppressed MHC I and IFNλ despite its efficient invasion capacity is not clear.

One of the most interesting observations was the presence of bacteria within ductal cells in the LSGs with FLS or nonspecific chronic inflammation. When the glandular dysfunction in SS had been understood as a result of acinar cell destruction by infiltrated cytotoxic T cells, the appearance of periductal, rather than periacinar, infiltration was a puzzle. Bacterial infection of ductal cells and its spread into neighboring acinar cells explains the pattern of lymphocytic infiltration in SS. Bacterial infection may also account for the previously reported constitutive high expression of TLR1, TLR2, TLR4, and TLR9 but not TLR3 in SGECs from SS patients [37, 38]. Bacterial infection of ductal cells also coincides with the fact that CXCL9, CXCL10, CXCL12, chemokines involved in lymphocyte trafficking, are predominantly expressed in the ductal epithelium adjacent to lymphoid infiltrates in SS [7, 8]. Furthermore, the infected ductal and acinar cells may provide a target to infiltrated T cells, leading to increased apoptosis observed in SS [39].

Since bacteria were detected within ductal cells and the area of infiltration with not only FLS but also nonspecific/sclerosing chronic inflammation, bacterial infection might be the

result of barrier disruption caused by inflammation and/or fibrosis. In such cases, bacterial infection would aggravate inflammation and deregulation of SGECs. However, even the ducts located in the areas without inflammation or fibrosis were infected with bacteria in the LSGs with FLS (S2 Fig). These results suggested the possibility that bacterial infection precedes FLS. Advanced FLS foci often form ectopic GC-like structures with segregated T and B cell areas, proliferating lymphocytes, and a network of follicular DCs [40]. The titers and the local production of anti-SSA and anti-SSB autoantibodies were significantly increased in patients with GC development [41]. The presence of bacteria in the GC-like structures may contribute to the proliferation of B cells and the production of autoantibodies, including anti-AQP5 autoantibodies [25], at local sites through direct stimulation of TLR4 and TLR9 or through exosomes produced from the infected cells. One unanswered question is how neutrophils are not recruited in the presence of bacteria. We speculate that the bacteria observed in the LSGs are mainly intracellularly located. Characterization of bacterial communities within the LSGs with FLS, localization of bacteria in relation with host cells, and further studies of host-microbe interactions would clarify the role of bacterial infection in SS.

## Conclusions

Dysbiotic oral microbiota may initiate the deregulation of SGECs and the IFN signature through bacterial invasion into ductal cells. In addition, bacterial infection of salivary glands may contribute to the perpetuation of inflammation and sialopathy in SS. These findings may provide new insights into the etiopathogenesis of SS.

## Supporting information

**S1 Fig. The specificity of Prevotella melaninogenica-specific probe.** The lysates of *Streptococcus salivarius* (Ss), *Neisseria oralis* (No), *Acinetobacter johnsonii* (Aj), *Streptococcus gordonii* (Sg), *Capnocytophaga gingivalis* (Cg), *Treponema denticola* (Td), *Fusobacterium nucleatum* (Fn), *Porphyromonas gingivalis* (Pg), *Saccharomonospora viridis* (Sv), and *Prevotella melaninogenica* (Pm) were prepared. After measuring the DNA concentration, each bacterial lysate that contains 100 ng DNA was blotted onto a nylon membrane. The membrane was then hybridized with the *P. melaninogenica*-specific probe.
(TIF)

**S2 Fig. The effect of IFN on expression of APC markers in HSG cells.** HSG cells ($4 \times 10^4$ cells/well) were plated into 24-well plates and treated with IFNγ for 72 hours. The expression of APC-related surface molecules on HSG cells was analyzed by flow cytometry and fluorescence microscopy. (A) Upregulation of APC markers such as MHC I, MHC II, and costimulatory molecules on HSG cells was analyzed by flow cytometry. (B) The expression of costimulatory molecules (green) on HSG cells in the absence or presence of IFNγ was confirmed by fluorescence microscopy.
(TIF)

**S3 Fig. Bacterial infection of ductal cells.** The sections of paraffin-embedded LSG tissues obtained from Control subjects or SS patients were subjected to H&E stain and *in situ* hybridization (ISH) using a universal (U) probe. (A) LSG with non-specific chronic inflammation from Control subjects. (B) LSG with FLS > 1 from SS patients. The areas marked with a square were examined with higher magnification. Arrows indicated infected ductal cells. Scale bars indicate 50 μm.
(TIF)

**S1 Table. The list of medications.**
(DOCX)

**S2 Table. Genera[a] associated with control vs. SS.**
(DOCX)

## Author Contributions

**Conceptualization:** Jung-Hyun Park, Youngnim Choi.

**Data curation:** Jehan Alam, Ahreum Lee, Junho Lee, Dong Il Kwon, Hee Kyung Park, Sumin Jeon, Jennifer Lee.

**Formal analysis:** Jehan Alam, Ahreum Lee, Junho Lee, Dong Il Kwon, Youngnim Choi.

**Funding acquisition:** Youngnim Choi.

**Investigation:** Hee Kyung Park, Jung-Hyun Park.

**Methodology:** Jehan Alam, Ahreum Lee, Junho Lee, Keumjin Baek.

**Resources:** Jennifer Lee, Sung-Hwan Park.

**Supervision:** Sung-Hwan Park, Youngnim Choi.

**Writing – original draft:** Jehan Alam.

**Writing – review & editing:** Ahreum Lee, Junho Lee, Dong Il Kwon, Hee Kyung Park, Jung-Hyun Park, Sumin Jeon, Keumjin Baek, Jennifer Lee, Sung-Hwan Park, Youngnim Choi.

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
