## [Decision Letter · Decision Letter 0]

13 Nov 2019

PONE-D-19-24479

Dysbiotic Oral Microbiota and Infected Salivary Glands in Sjögren’s Syndrome

PLOS ONE

Dear Professor Choi,

Thank you for submitting your manuscript to PLOS ONE. After careful consideration, we feel that it has merit but does not fully meet PLOS ONE’s publication criteria as it currently stands. Therefore, we invite you to submit a revised version of the manuscript that addresses the points raised by the reviewers during the review process.

We would appreciate receiving your revised manuscript by Dec 28 2019 11:59PM. To enhance the reproducibility of your results, we recommend that if applicable you deposit your laboratory protocols in protocols.io, where a protocol can be assigned its own identifier (DOI) such that it can be cited independently in the future. For instructions see: http://journals.plos.org/plosone/s/submission-guidelines#loc-laboratory-protocols

We look forward to receiving your revised manuscript.

Kind regards,

Silke Appel, PhD (Dr. rer. nat.)

Academic Editor

PLOS ONE

Journal Requirements:

Reviewers' comments:

Reviewer's Responses to Questions

**Comments to the Author**

1. Is the manuscript technically sound, and do the data support the conclusions?

Reviewer #1: Yes

Reviewer #2: Yes

2. Has the statistical analysis been performed appropriately and rigorously? 

Reviewer #1: Yes

Reviewer #2: No

3. Have the authors made all data underlying the findings in their manuscript fully available?

Reviewer #1: Yes

Reviewer #2: Yes

4. Is the manuscript presented in an intelligible fashion and written in standard English?

Reviewer #1: Yes

Reviewer #2: Yes

5. Review Comments to the Author

Reviewer #1: The manuscript entitled "Dysbiotic Oral Microbiota and Infected Salivary Glands in Sjogren's Syndrome" aims to characterize the oral microbiota in SS and describe the possible effects of selected oral pathogens on the immune response of the host. The authors have shown that several bacterial species differ significantly in abundance when comparing control subjects and Sjogren syndrome patients with and without dryness. Moreover, the authors the ability of the selected oral bacteria to induce the APC phenotype in HSG cells.

The article is well structured and well written. The methods are clearly described. The conclusions are appropriately reasoned. The findings, while not entirely novel, present excellent opportunity for further study, especially with regard to the sequence of events in the aetiology of Sjogren's syndrome. However, there are few minor issues that require adressing.

Minor comments

While the intestinal microbiota in SS patients and other disorders is an interesting avenue of research, it seems of no connection to the study describing only the oral microbiota. Especially since, in the case of diseases listed in line 66, oral microbiota is also being investigated, particularly with the connection to periodontitis.

It is unclear what the numbers in brackets in Tables 2-4 next to relative abunance of each bacteria species represent.

Some of the medications listed in S1Table are known to have bactericidal activity (such as simvastatin).

Throughout the manuscript, the authors state that "bacterial species was increased..." or "genera was decreased". Do the authors mean the abundance of a particular species?

The results presented in Fig. 2 were analysed with t-test. Why did the authors choose that as opposed to the one-way analysis of variance (ANOVA)?

It is unclear which studies the authors refer to when discussing "previous studies" in line 331 in the Discussion section. Salivary microbiota in SS has already been somewhat investigated and it could benefit the current manuscript to draw a more in-depth comparison with, for example, most recent reports (DOI):

10.1080/20002297.2019.1660566

10.1371/journal.pone.0218319 (referenced in the introduction of this manuscript)

Please clarify or modify the following sentences.

Line 80: All subjects were female because there were only four male SS patient in our cohort.

Line 97: All biopsies were from Asian female obtained at the time of diagnosis.

Line 334 - 336: The salivary microbiota of individuals with dental caries is characterized with a reduced Shannon diversity, which may be attributed to decreased salivary pH. Thus, the increased Shannon diversity int he previous studies may refelect increased caries prevalence in SS. [ref?]

Line 339: Interestingly, the shannon index had a weak but significant positive correlation with total bacterial load that was also significantly increased in SS.

Line 382: Since bacteria were detected within ductal cells and the area of infiltration with any type of inflammation, (...).

Please check abbreviations.

Reviewer #2: Alam et al reports dysbiotic oral microbiota in pSS. Beyond reporting dysbiosis, the authors have attempted to show role of microbiota in etiopathogenesis by demonstrating that P melaninogenica infected salivary gland epithelial cells and could upregulate MHC , costimulatory molecules and IFN in salivary gland epithelial cells.

Comments

1.In baseline characteristics ( Table 1) , please check if age of controls and ss without dryness and corresponding p value is correct.

2. Could the fold changes of microbiome between SS and controls also be provided?This would allow for comparison with previous studies

3.In contrast to at least 9 previous studies on oral microbiome in pSS, they have reported increase in diversity.In discussion, the authors state that this could be attributed to decrease in salivary gland defensins reported by Kaneda et al. However, proteomic studies have reported increase in antimicrobial peptides such as lactoferrin, lysozyme, defensin (Ryu

Et al Rheumatology 2016, Peluso et al Arthritis Rheum 2007

4.Also none of the previous studies have reported an increase in Prevotella. In fact two studies Siddiqui et al(Journal of microbiology 2016) and Rusthen et al (Plos one 2019) report increase Prevotella in controls.It is suggested that the authors discuss the prior microbiome studies in pSS.

Minor comments

Page 19 line 335 in place of increased Shannon Index , the authors have intended to mention decreased- possibly typo

6. PLOS authors have the option to publish the peer review history of their article (what does this mean?). If published, this will include your full peer review and any attached files.

Reviewer #1: No

Reviewer #2: No

---

## [Author Response · Author response to Decision Letter 0]

4 Jan 2020

We appreciate reviewers for their time and helpful comments that improved the quality of this manuscript.

Reviewer #1: 

Minor comments

While the intestinal microbiota in SS patients and other disorders is an interesting avenue of research, it seems of no connection to the study describing only the oral microbiota. Especially since, in the case of diseases listed in line 66, oral microbiota is also being investigated, particularly with the connection to periodontitis.

→ As the reviewer suggested, intestinal microbiota has been removed, and the sentence has been edited with an appropriate reference as following (lines 64-66): A growing body of evidence suggests that dysbiosis of the oral microbiome is associated with the pathogenesis of several autoimmune diseases, such as systemic lupus erythematosus, Crohn's disease, and rheumatoid arthritis. [9].

It is unclear what the numbers in brackets in Tables 2-4 next to relative abundance of each bacteria species represent.

→ Table legends have been added as following: aExpressed as the median (range)

Some of the medications listed in S1Table are known to have bactericidal activity (such as simvastatin).

→ Thank you for pointing out the antibacterial activity of non-antibiotic drugs. Frankly speaking, we did not know it. In Discussion section, this issue has been addressed as following (lines 371-377): It is noteworthy that antimicrobial activities have been observed by a number of non-antibiotic drugs such as the beta-adrenergic receptor antagonists, diuretic drugs, H1 antihistamines, psychotropic drugs, non-steroid anti-inflammatory drugs, and proton pump inhibitors [30,31], and some of the medications taken by the drug-related sicca patients are known to have antibacterial activity (S2 Table). Although the control subjects without or with dryness harbored comparable levels of total bacterial loads, the medications with antimicrobial activity may have affected the composition of oral microbiota. 

Throughout the manuscript, the authors state that "bacterial species was increased..." or "genera was decreased". Do the authors mean the abundance of a particular species?

→ “the relative abundance of” was put before the name of taxa, or “enriched” and “underrepresented” have been used instead of “increased” and “decreased”, respectively. 

The results presented in Fig. 2 were analysed with t-test. Why did the authors choose that as opposed to the one-way analysis of variance (ANOVA)?

→ Because we wanted to know if each species can induce the APC-like phenotype compared with non-infected control, not the comparison of the effects of different species.

It is unclear which studies the authors refer to when discussing "previous studies" in line 331 in the Discussion section. Salivary microbiota in SS has already been somewhat investigated and it could benefit the current manuscript to draw a more in-depth comparison with, for example, most recent reports (DOI): 10.1080/20002297. (referenced in the introduction of this manuscript)

→ Reference citation [10,11,13,14] has been added (line 337). In addition, a paragraph that compared the oral microbiota data with previous studies has been added (lines 348-356).

Please clarify or modify the following sentences.

Line 80: All subjects were female because there were only four male SS patient in our cohort.

→ All subjects were female because male SS patients are very rare in Korea. (Line 92)

Line 97: All biopsies were from Asian female obtained at the time of diagnosis.

→ We found out that participation to the SICCA study was not based on treatment or non-treatment. Therefore, “obtained at the time of diagnosis” has been removed from the sentence. (line 100)

Line 334 - 336: The salivary microbiota of individuals with dental caries is characterized with a reduced Shannon diversity, which may be attributed to decreased salivary pH. Thus, the increased Shannon diversity in the previous studies may reflect increased caries prevalence in SS. [ref?]

→ The sentence has been changed as following (line 340): Thus, the decreased Shannon diversity in the previous studies may reflect increased caries prevalence in SS.

Line 339: Interestingly, the shannon index had a weak but significant positive correlation with total bacterial load that was also significantly increased in SS.

→ Line 345: Interestingly, the Shannon index had a weak but significant positive correlation with total bacterial load that was also significantly increased in SS compared to control.

Line 382: Since bacteria were detected within ductal cells and the area of infiltration with any type of inflammation, (...).

→ Line 404-405: Since bacteria were detected within ductal cells and the area of infiltration with not only FLS but also nonspecific/sclerosing chronic inflammation, bacterial infection might be the result of barrier disruption caused by inflammation and/or fibrosis.

Please check abbreviations.

→ Editorial errors regarding the use of abbreviations have been corrected. 

Reviewer #2 Comments: 

1.In baseline characteristics (Table 1) , please check if age of controls and ss without dryness and corresponding p value is correct.

→ Yes, it’s correct by ANOVA with Bonferroni post hoc test.

2. Could the fold changes of microbiome between SS and controls also be provided? This would allow for comparison with previous studies

→ Statistically differently distributed taxa between SS (n=25) and controls (n=25) are already presented at the phylum (Fig. 1E), genera (S2 Table), and species levels (Table 2). Because the relative abundance of SS (n=25) and controls (n=25) groups for each genera are not shown in the S2 Table, the names of 5 genera enriched in SS have been listed in the result section line 195 as following (line 195): At the genus level, 5 genera, including Streptococcus, Prevotella, Lactobacillus, Atopobium, and Staphylococcus, were enriched but 34 genera were underrepresented in SS compared to control, among which 9 and 8 genera maintained significant differences in non-dry and dry conditions, respectively (S2 Table).

3.In contrast to at least 9 previous studies on oral microbiome in pSS, they have reported increase in diversity. In discussion, the authors state that this could be attributed to decrease in salivary gland defensins reported by Kaneda et al. However, proteomic studies have reported increase in antimicrobial peptides such as lactoferrin, lysozyme, defensin (Ryu et al Rheumatology 2016, Peluso et al Arthritis Rheum 2007)

→ Proteomics studies used the same volume of saliva for analysis. The majority of differently expressed proteins are up-regulated in SS patients (Cecchettini et al. Clinical Proteomics 2019;16:26), but this, in a way, reflects concentration of proteins due to the reduced salivary flow rate in SS patients. Furthermore, many PRPs, statherin, and Histatins that have a role in agglutination of bacteria and generation of antimicrobial peptides were decreased in the saliva of SS (Peluso et al Arthritis Rheum 2007). Therefore, we left the citation of decreased expression of human beta-defensins in the salivary gland of SS patients.

4. Also none of the previous studies have reported an increase in Prevotella. In fact two studies Siddiqui et al(Journal of microbiology 2016) and Rusthen et al (Plos one 2019) report increase Prevotella in controls. It is suggested that the authors discuss the prior microbiome studies in pSS.

→ Considering the differences in the sampling sites (whole mouthwash, saliva, tongue, and buccal mucosa), ethnicity, diet, and the sequenced V regions of 16S rRNA gene, the results of oral microbiota analysis cannot be directly compared with those of previous studies. For example, Meulen et al. [14] reported lower Streptococcus relative abundances in pSS compared with HCs, while ref 11-13 and our results showed higher oral Streptococcus relative abundance in patients with pSS compared with HCs. One previous study by Li et al [12] reported increased Prevotella in SS. The increased relative abundance of Prevotella in controls reported by Siddiqui et al(Journal of microbiology 2016) and Rusthen et al (Plos one 2019) were not significantly different from those in SS. We added a paragraph in Discussion section that addressed the changes in the bacterial composition in comparison with previous works as following: Considering the differences in the sampling sites (whole mouthwash, saliva, tongue, and buccal mucosa), ethnicity, diet, and the sequenced V regions of 16S rRNA gene, the results of oral microbiota analysis cannot be directly compared with those of previous studies. Nevertheless, there are several common findings. Increased relative abundances of phylum Firmicutes [10,13] and genera Streptococcus [11-13], Lactobacillus [10,12], Prevotella [12], and Atopobium [10] but decreased relative abundances of phyla Proteobacteria [10,12] and Spirochaetes [13] and genera Neisseria [10,12], Haemophilus [10,12], Leptotrichia [11,12], Fusobacterium [11], Lautropia [10], Porphyromonas [12,13], Treponema [13], and Tannerella [12,13] in SS compared to those in controls have been repeatedly observed, including within the current study. The reduced relative abundance of P. gingivalis also agrees with previous studies [11, 13] and explains the lack of increased risk of periodontitis in SS patients, despite the reduced salivary antimicrobial function [28, 29].

Minor comments

Page 19 line 335 in place of increased Shannon Index , the authors have intended to mention decreased- possibly typo

→ Thanks for pointing out this mistake. It has been corrected.

---

## [Decision Letter · Decision Letter 1]

6 Feb 2020

PONE-D-19-24479R1

Dysbiotic Oral Microbiota and Infected Salivary Glands in Sjögren’s Syndrome

PLOS ONE

Dear Professor Choi,

Thank you for submitting your manuscript to PLOS ONE. After careful consideration, we feel that it has merit but does not fully meet PLOS ONE’s publication criteria as it currently stands. Therefore, we invite you to submit a revised version of the manuscript that addresses the points raised during the review process.

We would appreciate receiving your revised manuscript by Mar 22 2020 11:59PM. To enhance the reproducibility of your results, we recommend that if applicable you deposit your laboratory protocols in protocols.io, where a protocol can be assigned its own identifier (DOI) such that it can be cited independently in the future. For instructions see: http://journals.plos.org/plosone/s/submission-guidelines#loc-laboratory-protocols

We look forward to receiving your revised manuscript.

Kind regards,

Silke Appel, PhD (Dr. rer. nat.)

Academic Editor

PLOS ONE

Reviewers' comments:

Reviewer's Responses to Questions

**Comments to the Author**

1. If the authors have adequately addressed your comments raised in a previous round of review and you feel that this manuscript is now acceptable for publication, you may indicate that here to bypass the “Comments to the Author” section, enter your conflict of interest statement in the “Confidential to Editor” section, and submit your "Accept" recommendation.

Reviewer #1: All comments have been addressed

Reviewer #2: (No Response)

2. Is the manuscript technically sound, and do the data support the conclusions?

Reviewer #1: Yes

Reviewer #2: Yes

3. Has the statistical analysis been performed appropriately and rigorously? 

Reviewer #1: Yes

Reviewer #2: Yes

4. Have the authors made all data underlying the findings in their manuscript fully available?

Reviewer #1: Yes

Reviewer #2: Yes

5. Is the manuscript presented in an intelligible fashion and written in standard English?

Reviewer #1: Yes

Reviewer #2: Yes

6. Review Comments to the Author

Reviewer #1: (No Response)

Reviewer #2: A number of recent papers on Sjogren's saliva microbiome which appeared in the recent years have not been cited.

7. PLOS authors have the option to publish the peer review history of their article (what does this mean?). If published, this will include your full peer review and any attached files.

Reviewer #1: No

Reviewer #2: No

---

## [Author Response · Author response to Decision Letter 1]

10 Feb 2020

We appreciate reviewers for their time and helpful comments that improved the quality of this manuscript.

Reviewer #2: A number of recent papers on Sjogren's saliva microbiome which appeared in the recent years have not been cited.

→ The two papers below published in 2019 were additionally cited in Introduction (Page 4 line 1-3) and Discussion (Page 19 line 20) sections.

16. Sharma D, Sandhya P, Vellarikkal SK, Surin AK, Jayarajan R, Verma A, et al. Saliva microbiome in primary Sjogren's syndrome reveals distinct set of disease-associated microbes. Oral diseases. 2019. Epub 2019/09/13. doi: 10.1111/odi.13191. PubMed PMID: 31514257.

17. Sembler-Møller ML, Belstrøm D, Locht H, Enevold C, Pedersen AML. Next-generation sequencing of whole saliva from patients with primary Sjögren’s syndrome and non-Sjögren’s sicca reveals comparable salivary microbiota. Journal of Oral Microbiology. 2019;11(1):1660566. doi: 10.1080/20002297.2019.1660566.

---

## [Editor Report · Decision Letter 2]

6 Mar 2020

Dysbiotic Oral Microbiota and Infected Salivary Glands in Sjögren’s Syndrome

PONE-D-19-24479R2

Dear Dr. Choi,

We are pleased to inform you that your manuscript has been judged scientifically suitable for publication and will be formally accepted for publication once it complies with all outstanding technical requirements.

With kind regards,

Silke Appel, PhD (Dr. rer. nat.)

Academic Editor

PLOS ONE
---

## [Editor Report · Acceptance letter]

10 Mar 2020

PONE-D-19-24479R2 

Dysbiotic Oral Microbiota and Infected Salivary Glands in Sjögren’s Syndrome 

Dear Dr. Choi:

I am pleased to inform you that your manuscript has been deemed suitable for publication in PLOS ONE. Congratulations! Your manuscript is now with our production department. 

With kind regards,

on behalf of

Dr. Silke Appel 

Academic Editor

PLOS ONE